# Control of nuclear dynamics in the benzene cation by electronic wavepacket composition

Thierry Tran [1,2✉], Graham A. Worth [1✉] & Michael A. Robb [2✉]

The study of coupled electron-nuclear dynamics driven by coherent superpositions of electronic states is now possible in attosecond science experiments. The objective is to understand the electronic control of chemical reactivity. In this work we report coherent 8-state non-adiabatic electron-nuclear dynamics simulations of the benzene radical cation. The computations were inspired by the extreme ultraviolet (XUV) experimental results in which all 8 electronic states were prepared with significant population. Our objective was to study the nuclear dynamics using various bespoke coherent electronic state superpositions as initial conditions in the Quantum-Ehrenfest method. The original XUV measurements were supported by Multi-configuration time-dependent Hartree (MCTDH) simulations, which suggested a model of successive passage through conical intersections. The present computations support a complementary model where non-adiabatic events are seen far from a conical intersection and are controlled by electron dynamics involving non-adjacent adiabatic states. It proves to be possible to identify two superpositions that can be linked with two possible fragmentation paths.

[1] Department of Chemistry, University College London, London, UK. [2] Department of Chemistry, Molecular Sciences Research Hub, Imperial College London, London, UK. ✉email: thierry.tran.18@ucl.ac.uk; g.a.worth@ucl.ac.uk; mike.robb@imperial.ac.uk

It is well known that the Born–Oppenheimer approximation breaks down near conical intersections[1–7] where electronic and nuclear motion become highly coupled. In photochemistry, a mechanism is often formulated in terms of several sequential steps involving (i) motion on an (excited) adiabatic state followed by (ii) non-adiabatic dynamics at a conical intersection and (iii) motion on an (ground) adiabatic state. In contrast, it is possible to start photochemical dynamics with a coherent superposition of adiabatic electronic states created in a laser experiment[8–11]. In this paper, we demonstrate that for the case of a coherent superposition of adiabatic states, non-adiabatic effects can occur far from a conical intersection and involve nonadjacent adiabatic states. We illustrate this idea with the eight lowest energy states of the benzene cation, which has been the subject of recent experimental work[12] using extremely short extreme ultraviolet (XUV) pulses obtained by means of high-order harmonic generation (HHG). In addition, theoretical dynamics studies were reported using the MCTDH method[13]. The central conclusion of Galbraith et al.[12] was that the main experimental observations could be understood by a scheme involving successive E→D and D→B conical intersections (see Fig. 1a). These state labels will be discussed subsequently and are given in Supplementary Note 1 and Fig. 2. We use two notations for the electronic states (i) historic spectra notation ($E_8$, $D_7$, $D_6$), where we use subscripts to remind ourselves of the energy ordering, and (ii) the irreducible representation label ($E_{1u}$, $B_{2u}$, etc.).

In earlier work, much of the effort in this field was focused on electron dynamics with fixed nuclei[14–16]. However, studies of the effect of nuclear motion on this electron dynamics[17–19] showed that it could not be neglected. The effect of electronic state mixing at a conical intersection has also been shown for the two lowest cation states of benzene[20]. In this work, we consider the ideal situation where the lowest eight states are excited coherently with weights derived from photoelectron cross sections. These eight states were studied in the Galbraith[12] experiments but it is not possible to know the extent to which they were coherently populated. Thus, while direct comparison with experiment is not possible, we believe this study uncovers some general principles for the interpretation of XUV induced dynamics.

The overall mechanistic scenario obtained with a model involving successive decay through conical intersections is shown in Fig. 1a[12]. The system descends from state E through the E/D conical intersection, etc. At the conical intersection, the symmetry can break due to derivative couplings driving the nuclear dynamics.

The model used in this paper, which involves the superposition of adiabatic states is illustrated in Fig. 1b for state E and one component of the B state. A detailed description of the electronic states will come in later discussion. At this stage, we want merely to indicate the main ideas. The important idea is that states E and B are non-degenerate (Fig. 1a). In Fig. 1b, we show the Z-axis, in the direction of the derivative coupling (interstate gradient). The X and Y axes represent the electronic structure of state E (X-axis) and a component of state B (Y-axis) as well as the two normal modes with the same symmetry nm "e" and nm "b". The σ orbital electronic states are represented in Fig. 1b as "bond orbitals". In this way, they transform in the same way as the C–C stretches and they have a simple analytic representation. The states E and B are stationary states and at the zero of the Z-axis, there are no symmetry breaking gradients in the X–Y plane. The initial oscillations of the electron wavepacket thus result in a potential gradient in the direction of the derivative couplings, i.e., along the Z-axis. The initial nuclear motion is thus driven in this direction. As the system moves along the Z-direction, from zero time, gradients develop in the XY plane because symmetry is broken.

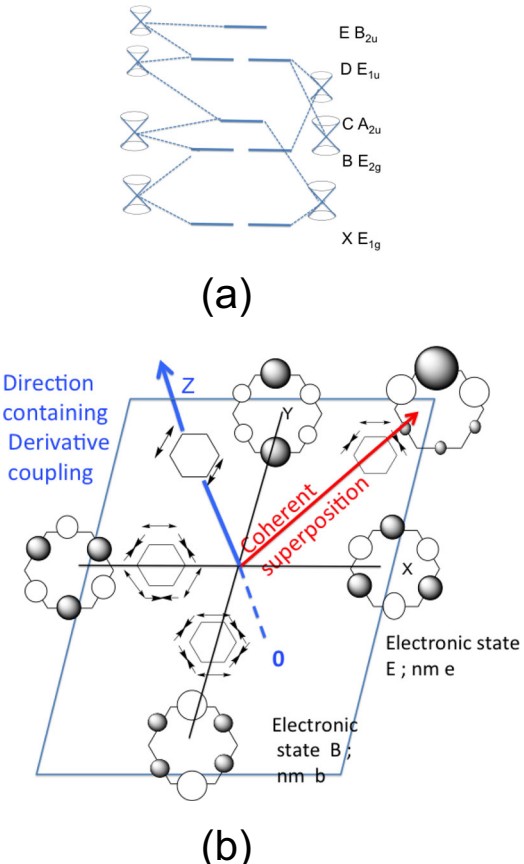

**Fig. 1 Comparison of non-adiabatic dynamics mechanisms. a** From successive decay through conical intersections E→D→B→ etc. versus **b** from a coherent superposition of two nonadjacent electronic states E and B. In **b**, the electronic states are represented with open (negative) and shaded circles (positive) representing the lobes on the electron density in the hole state MO. The normal mode stretches are represented with arrows. A more detailed discussion of the representation of the electronic states will be given subsequently. The Z-axis represents the interstate gradient as a combination of C–C stretches. At 0 on the Z-axis, there are no components of the gradient that do not belong to non-totally symmetric representations. As one moves along the Z-axis, one breaks symmetry and nonsymmetric gradients that develop in the XY plane, and one sees electron and nuclear dynamics along the red axis.

The red axis represents the coherent superposition of states E and one of the components of B. Electron dynamics occurs along this (red) direction in concert with the nuclear dynamics along a corresponding superposition of normal modes. Thus for the successive decay model (Fig. 1a), the transition through the conical intersection is associated with a two-dimensional branching plane (where the degeneracy is lifted) at the conical intersection. One of these directions involves the derivative coupling (interstate gradient). In contrast in Fig. 1b, for states that are not degenerate, the component of the gradient that is non-totally symmetric lies along the derivative coupling (Z-axis in Fig. 1b). Because we have a superposition of the states associated with X and Y axes, we have electron dynamics and coupled nuclear dynamics along the red vector. In the special case where the states along the X and Y axes are degenerate in Fig. 1b, we need only the XZ or ZY axes and the X or Y axis can be chosen as a linear combination and the electron dynamics takes place in this plane (i.e., the conventional conical intersection picture).

Our strategy in this paper involves the study of non-adiabatic dynamics of a coherent superposition of the electronic states of

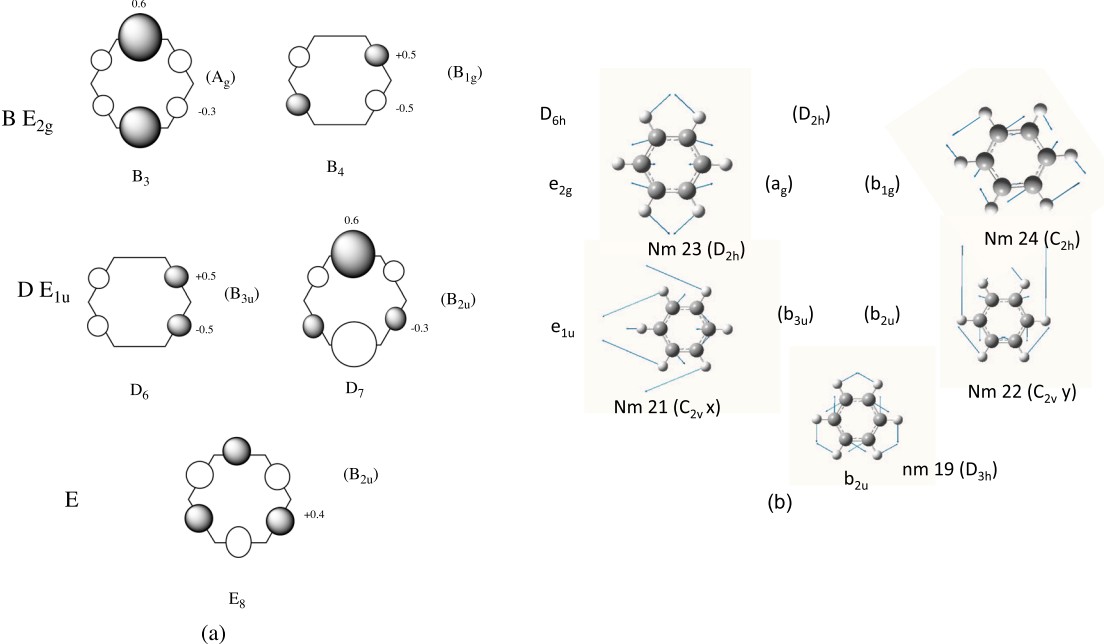

**Fig. 2 Reference diabatic states (time 0) and normal modes. a** Diabatic states (at time 0) in the permutation representation (Eq. (3)). E D B are spectroscopic notation (the $\pi$ states are shown with orbital plots in SI), $E_{1u}$, $E_{2g}$, etc. refer to $D_{6h}$ symmetry while the additional labels in brackets ($B_{2u}$) .... refer to $D_{2h}$ symmetry. Each shaded lobe represents a positive contribution while an unshaded lobe is negative. The coefficient for each lobe is from Eq. (3). **b** Computed normal modes spanned by the same permutation representation. They are ordered from lowest energy to highest. The nature of the symmetry lowering for each normal mode is indicated in parenthesis. The couplings of the electronic states and normal modes are indicated in Table 1 and Supplementary Note 4 and Figure S2.

benzene. This is motivated by the work of Galbraith[12]. But practically more general concepts will emerge. Thus we will not attempt to compare with the Galbraith[12] experiments except in a general way.

The central conclusion from this work is the demonstration of a decay mechanism which is an alternative to involving successive decay through conical intersections. We have shown that the non-adiabatic effects (e.g., coupled electron-nuclear dynamics) associated with coherent superpositions can be seen far from the conical intersection and may be partly responsible for fragmentation.

**Computational and theoretical background**. We now discuss some important aspects of the theory and practical details. A more complete discussion of the initial conditions for the dynamics and the Qu–Eh algorithm is given in Supplementary Note 1 and 2.

The non-adiabatic dynamics computations to be discussed in this paper were performed using the Quantum-Ehrenfest (Qu–Eh) method[21] which combines a CAS-CI formulation of the Ehrenfest method for the electronic motion[22] and the direct dynamics variational multi-configuration Gaussian (DD-vMCG) algorithm for nuclear dynamics[23,24]. Thus the initial electronic structure can be chosen to be a coherent superposition of adiabatic states that is propagated as a solution of the time-dependent Schroedinger equation. We choose both an 8-state coherent superposition (within a sudden approximation), and various bespoke coherent superpositions designed to unravel the origins of various non-adiabatic effects.

An important feature of the electronic structure part of the Qu–Eh method[21] is that the full derivative coupling is included in the expression for the analytic gradient[22]. Thus there are off-diagonal terms between adiabatic states occurring in the superposition (called derivative couplings in other contexts) that are included. We shall demonstrate that these off-diagonal terms lead to non-adiabatic motion, of a type normally seen at a conical intersection, but, in this case, occurs between states that are non-degenerate and far from a conical intersection.

In the Qu–Eh method, the electronic motion is described by a CAS-CI wavefunction that is propagated as a solution of the time-dependent electronic Schroedinger equation using the Ehrenfest method as discussed in Vacher et al.[22] implemented within a development version of Gaussian[25]. Here we used a CAS space with 15 electrons and 8 orbitals. The orbitals were taken from an 8-state CASSCF at the symmetric $D_{6h}$ minimum geometry. The orbitals are propagated to second order in the orbital rotation and re-orthogonalization parameters.

The nuclear motion is also fully quantum. The nuclear dynamics is propagated using methods implemented in Quantics[24] which use Gaussian wavepackets (gwp). Each initially unpopulated gwp is associated with an "excitation" of a normal mode from the ground-state neutral wavefunction (for a detailed discussion of the initial conditions see Supplementary Notes 1 and 2). In our case, we used the 12 normal modes that described the in-plane C–C stretches and C–C–C angle bends. Thus we have $2 \times 12 + 1$ gwp in our computations. (A single 61-gwp 8-state computation was allowed to run for a short time and the results were very similar to the 25 gwp computations.) In order to reduce errors in the integration, we used a width of 0.1 rather than 0.707 (the width of the ground-state vibrational wavefunction in the harmonic approximation) in the definition of the gwp. We used a time-step of 0.1 fs with 5th order Runge–Kutta integration. The normal modes were obtained from a 6–31g* B3LYP (Becke, 3-parameter, Lee–Yang–Parr) computation at the neutral equilibrium geometry. The initial distribution of the gwp (see Supplementary Note 2 for details) was made in momentum space so all gwp were started at the same geometry.

The degenerate orbitals and normal modes were chosen to belong to irreducible representations of $D_{2h}$ in addition to $D_{6h}$. Thus, for example, the $E_{2g}$ representation of $D_{6h}$ when restricted to $D_{2h}$ gives $A_g$ and $B_{1g}$. So the degenerate orbitals and vibrations are adapted through this subgroup chain. This choice is not unique but involves no loss in generality.

The main results of the computations are presented as the displacement along the normal modes and reference diabatic states (definition to be discussed subsequently). For the normal modes, all expectation values were evaluated by averaging over the 25 gwp using the gross Gaussian population (GGP[26]). We also present the populations of diabatic states. In our computations, these correspond to the weights of the configuration state functions that correspond to the adiabatic states (those that diagonalize the CI Hamiltonian) at time zero. Since the orbitals in these configuration state functions change only very slowly with time (mainly due to re-orthogonalization), this is a suitable reference for the electron dynamics. The diabatic states are also averaged using gross populations. These diabatic states (at time zero) are given in Supplementary Note 3 Figure S1. We shall present these in a simpler form subsequently (Fig. 2a).

In the initial superposition of adiabatic states that is propagated, we have chosen all the weights to be positive. For a pair of states, changing the phase, merely drives the electron dynamics in one direction rather than the other. Further, the overall phase of both the adiabatic states and the normal modes is arbitrary. The plots in Figure S1 in Supplementary Note 3 and Fig. 2 (to be discussed subsequently) show the phases actually used. In the subsequent discussion, we will discuss a possible fragmentation pattern. Changing these phases and/or the phase of the adiabatic states and normal modes themselves produces additional superpositions that are not considered in this manuscript.

## Results and discussion
**Symmetry, diabatic states, normal modes, and superpositions.**
As we shall discuss in detail, the nuclear and electron dynamics are controlled by two effects (see Fig. 1b and related discussion). At time zero there is a component of the gradient that breaks symmetry. Then, because we are mixing two or more adiabatic states, we will obtain an oscillating mixture of these states as a function of time. We shall refer to this as electron dynamics. Because of the high symmetry of Benzene (see Doscher et al.[27] or Galbraith et al.[12] for a full discussion), much electronic structure and related dynamics is controlled by symmetry effects. To simplify this discussion it is convenient to introduce the reducible permutation representation (see the tables by Atkins[28]). The basis $(h_1, h_2 \ldots h_6)$ for this representation is the set of C–C bonds and/ or the set of C–C stretches. Via this representation, it becomes possible to relate the electronic state symmetry and the C–C stretching symmetry in the electron dynamics.

We begin with a discussion of the potential gradient due to a superposition of electronic states at time zero. The gradient (or force) that drives the non-adiabatic dynamics of a coherent superposition has two types of components: intrastate and interstate. The latter (off-diagonal gradients that arise from the mixing) are the derivative couplings and have the form $\langle \psi_I | \partial/\partial Q_i^{\alpha_i} \hat{H}_e | \psi_{II} \rangle$ where the I and II are two adiabatic states and $\partial/\partial Q_i^{\alpha_i} \hat{H}_e$ is the gradient operator for each normal mode $i$ with symmetry $\alpha_i$. The gradient of the intrastate terms is only non zero along normal modes belonging to totally symmetric irreducible representations (except for a Jahn–Teller conical intersection[29] for the Jahn–Teller active modes). For the interstate terms, the mixing/superposition of two states I and II will be "allowed", with an off-diagonal gradient component along $Q_i^{\alpha_i}$,

**Table 1 Symmetry couplings for examples presented numerically in this work (see Supplementary Note 4 Figure S2 for full tabulation).**

| States | nm from electronic coupling | nm from electron dynamics |
|---|---|---|
| $E_8$ $B_3$ | 19 22 | 19 23 |
| $E_8$ $B_4$ | 21 | 19 24 |
| $E_8$ $D_6/D_7$ | 24/23 | 19 21/19 22 |
| $B_3$ $B_4$ | 24 | 23 24 |
| $D_6$ $D_7$ | 24 | 21 22 |

only if $\alpha^I \otimes \alpha^{Q_i^{\alpha_i}} \otimes \alpha^{II} = A_{1g}$ ($A_{1g}$ is the totally symmetric representation) and $\alpha^I$ in an irrep. label). Because of the high symmetry of benzene, these selection rules are very rich and are collected in Figure S2 in Supplementary Note 4 for both the normal modes of neutral benzene and the eight states considered in the benzene cation. We will discuss individual superpositions (Table 1) in more detail subsequently.

Now we turn to the electron dynamics. We consider, as an example, the superposition of the two states 1 and 2 with the wavefunction given in Eq. (1) below.

$$\Psi = z_1(t)e^{-iE_1(t)t}\phi_1(t) + z_2(t)e^{-iE_2(t)t}\phi_2(t) \qquad (1)$$

$$|\Psi(t)|^2 = |z_1(t)|^2|\phi_1(t)|^2 + |z_2(t)|^2|\phi_2(t)|^2 \\ + 2\Re\left(z_1(t)^*z_2(t)e^{i(E_1(t)-E_2(t))t}\phi_1(t)^*\phi_2(t)\right) \qquad (2)$$

From Eqs. (1) and (2) it is clear that one will see oscillations (electron dynamics) if two eigenstates are both populated and the larger the energy difference is, the shorter the period of the oscillations will be.

The important point about the electron dynamics and symmetry involves the connection between electron and nuclear dynamics (a simple version was illustrated in previous work[30] and we have introduced the ideas in Fig. 1b). The superposition of two electronic eigenstates, A (with symmetry a) and B (with symmetry b) yields oscillatory electron dynamics between A and B. The electron dynamics "nudges" the corresponding nuclear dynamics. In the case of the permutation representation, to be discussed next, this implies that the coupled (partly synchronous) electron-nuclear motion will involve states with symmetry a and b.

We now discuss the symmetry aspects of the electron dynamics and its effect on nuclear motion using the permutation representation. The C–C sigma bonds and the C–C stretches span a reducible permutation representation $(h_1, h_2 \ldots h_6)$ of $D_{6h}$. The symbols $(h_1, h_2 \ldots h_6)$ label the six C–C stretches or a set of six localized bond orbitals. The localized bond orbitals might be those formed by an "s" orbital at the center of the bond. These would reproduce the symmetry and nodal properties of the MO to be found in Supplementary Note 3 Figure S1. These basis functions are permuted under the symmetry operations of $D_{6h}$. The irreducible representations correspond to $A_{1g}$, $B_{2u}$, $E_{2g}$, and $E_{1u}$. The basis functions for the degenerate representations can be chosen within $D_{2h}$ as $A_g$ and $B_{1g}$ for $E_{2g}$ and $B_{3u}$ and $B_{2u}$ for $E_{1u}$ where we choose $\sigma_h \rightarrow \sigma_{yx} \sigma_v \rightarrow \sigma_{yz}$. One could make many choices of subgroup. However, as we shall discuss subsequently, one of the experimental fragmentation pathways observed are (C–C–C) + (C–C–C) fragments so $D_{2h}$ is a convenient choice. With this choice, the symmetry adapted (unnormalzed) linear

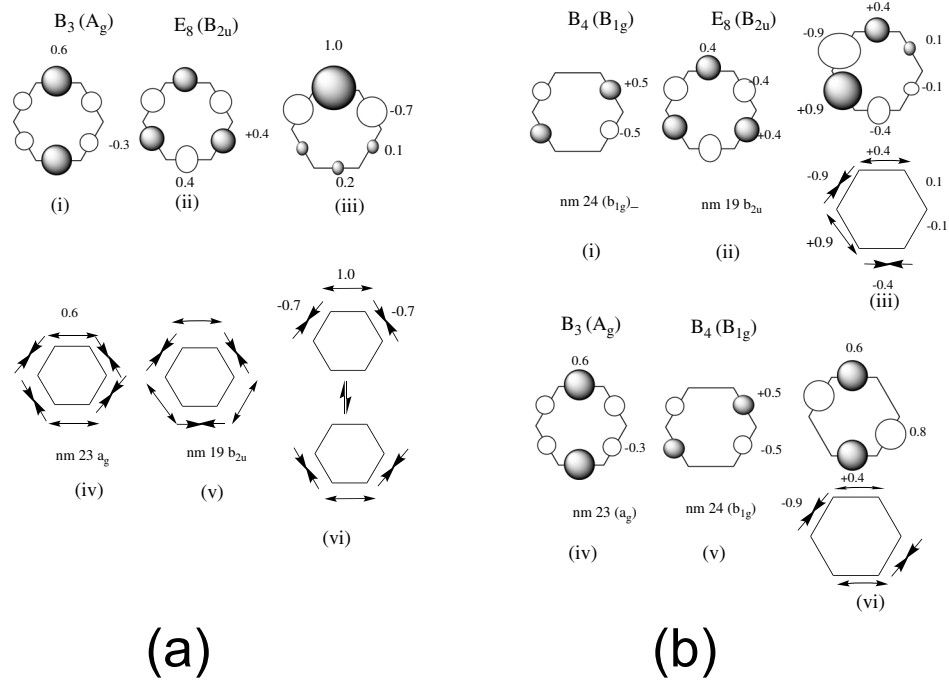

**Fig. 3 Examples of superpositions of electronic states and normal modes. a** The superposition of a component of the B state ($B_3$, part i) which is a component of the B state (with symmetry $E_{2g}$ in $D_{6h}$ and $A_g$ in $D_{2h}$), mixed with $E_8$ (part ii) (with symmetry $B_{2u}$) and a similar relation for nm 23 (part iv), and nm 19 (part v). **b** As for **a**, but mixing $B_4$ $E_8$ (parts i and ii) and $B_3$ $B_4$ (parts iv and v).

combinations are given[28] as

$$X_{A_{1g}} = h_1 + h_2 + h_3 + h_4 + h_5 + h_6$$
$$X_{B_{2u}} = h_1 - h_2 + h_3 - h_4 + h_5 - h_6$$
$$X_{E1u(B2u)} = 2h_1 - h_2 + h_3 - 2h_4 + h_5 - h_6$$
$$X_{E1u(B3u)} = h_2 + h_3 - h_5 - h_6 \qquad (3)$$
$$X_{E2g(Ag)} = 2h_1 - h_2 - h_3 + 2h_4 - h_5 - h_6$$
$$X_{E2g(B1g)} = h_2 - h_3 + h_5 - h_6$$

The diabatic states (hole states at time zero) for the CC bonding orbitals are collected in Fig. 2a. (The full electron density plots are given in Supplementary Note 3 Figure S1 for all states including the π states C and X.) Note that in this figure, we give the spectroscopic notation E, D, C.…X, the symmetry of the adiabatic state, $B_{2u}$, $E_{1u}$, etc., and the symmetry of the state in $D_{2h}$ in brackets. In Fig. 2a, the closed circles correspond to positive and the open circles to negative electron density. These give the same pattern of nodes as the plots in Supplementary Note 3 Figure S1 without any internal nodes. The numerical annotation gives the (normalized) weight from Eq. (3). The computed normal modes are shown in Fig. 2b. Here one can see the 1:1 correspondence with Eq. (3). The important point is that the representation in Eq. (3) is analytic. The states or vibrations can be combined algebraically and then displayed pictorially as in Fig. 2a or Fig. 1b.

We now discuss some examples (shown in Fig. 3), in addition to the introductory discussion in Fig. 1b, that will be directly relevant to our subsequent development. Let us consider (Fig. 3a) the superposition of a component (part i of Fig. 3a) of the B state $B_3$ (with symmetry $E_{2g}$ in $D_{6h}$ and $A_g$ in $D_{2h}$) mixed with $E_8$ (with symmetry $B_{2u}$, part ii of Fig. 3a). The positive combination gives part iii (Fig. 3a). The corresponding normal modes 23 (Fig. 3a part iv) and 19 (Fig. 3a part v) can be added to give the resultant vibration that breaks symmetry to $C_{2v}$. The direct product of $B_{2u}$ and $A_g$ is $B_{2u}$ corresponding to nm 19 and 22. Thus in the

superposition of E and $B_3$ we should see modes 19 and 22 from the electronic coupling and 19 and 23 from the electron dynamics. Looking at the model scheme in Fig. 1b, the Z-axis would contain modes 19 and 22, while the X and Y axes correspond to modes 19 and 23.

In Fig. 3b we show the corresponding analysis from $B_4$ and E and $B_3$ and $B_4$ and the results for these couplings and electron dynamics as well as D–D are collected in Table 1. Notice that for E–B mixing the resulting vibrations that break symmetry to $C_{2v}$, the symmetry of (C–C) + (C–C–C–C) fragments while B–B has $C_{2h}$ symmetry corresponding to (C–C–C) + (C–C–C) fragments.

**Eight-state non-adiabatic dynamics.** We begin with a discussion of our results from Quantum-Ehrenfest dynamics obtained with the full coherent superposition of 8 states. In our computations, we have weighted 8 states using the photoelectron cross sections of the states taken from Fig. 2 in the Galbraith experiments[12]. The values used in our computations were $X_1$ 0.051, $X_2$ 0.051, $B_3$ 0.179, $B_4$ 0.179, $C_5$ 0.449, $D_6$ 0.028, $D_7$ 0.028, and $E_8$ 0.0311, where the labels are defined in Fig. 2a (and Supplementary Note 3 Fig. 1). The dynamics was started with these weights (Fig. 4a) at time zero. (We have omitted, from Fig. 2a, 3 states that do not couple significantly for clarity). In the experiment, derived from the 8-state superposition, one observes fragmentation into two (C–C–C) fragments and (C–C) + (C–C–C–C) fragments. In our computations using the full 8-state coherent superposition, we can identify two different types of coherent C–C bond oscillations ($C_1$–$C_6$ + $C_3$–$C_4$) and ($C_1$–$C_2$ + $C_5$–$C_6$) as shown in Fig. 4b and c. The subscripts on the C atoms start with 1 at the top and increase clockwise around the ring. Note that we have taken all the weights of the adiabatic states to be positive and we could generate other solutions by changing the relative signs of the mixing. The fragmentation takes place on a long timescale. It is controlled, both by an orientation effect where the vibrational energy is deposited and by the energy barriers to fragmentation. We address only the first factor and assume it is related to the

 5

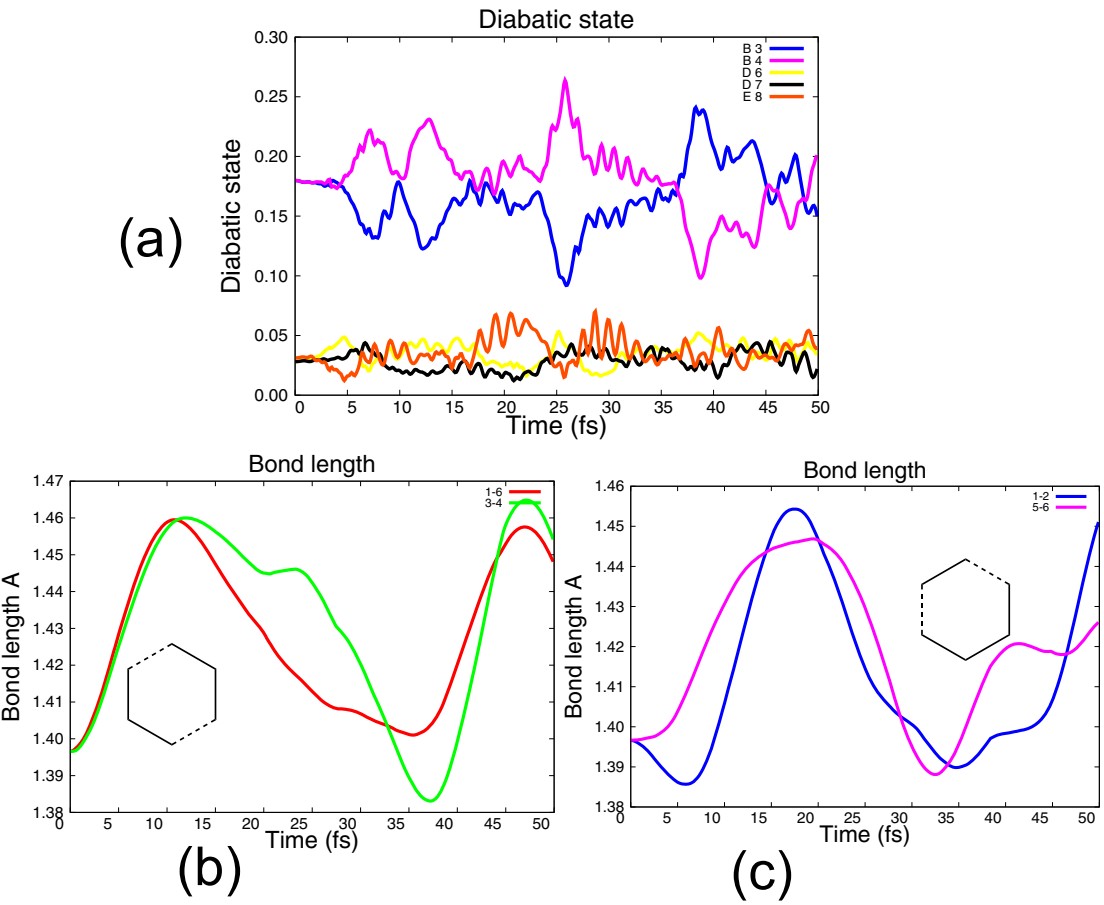

**Fig. 4 Results of 8-state Qu–Eh dynamics using the coherent superposition obtained from photoelectron cross sections.** The labels $E_8$, $D_7$, and $D_6$ refer to the singly ionized diabatic states according to Fig. 2a in Galbraith et al.[12] (and Figure S1). The atom labels 1–6 on the carbon atoms start at the top of the hexagon and increment clockwise. **a** Populations of the diabatic states. **b**, **c** Bond vibrations for dominant stretches. The data in all panels are obtained by weighting the gwp by gross populations[26].

computed initial vibrational distribution. Thus it would appear that we can see the two orientation effects in Fig. 4b and c.

The diabatic states (time 0) and their symmetries are shown in Fig. 2a. It now remains to understand the mechanism, i.e., which components of the superposition lead to the two observed fragmentation effects. Accordingly, we now report some additional simulations where we couple only selected states to try to explain the origin of the (C–C–C) + (C–C–C) and (C–C) + (C–C–C–C) fragment pathways. Looking at Fig. 4a, we can see that the dynamics should be dominated by the degenerate state B ($B_3$ $B_4$) and the mixing of the B state with the state E and the states D ($D_3$ $D_4$). We have investigated all possible bespoke mixings in our computations, including complex rotations but we focus on the E, D, and B states where the main mechanistic insight is given. We should emphasize at this point that we have created the 8-state superposition in a sudden approximation and we have not attempted to include the effect of the rise and fall of the HHG pulse components with time. However, for the bespoke superpositions, which we now discuss, we look at the specific mechanistic effect: different combinations of electronic wave-packets drive different nuclear motions.

**Non-adiabatic dynamics of E, E+D, E+B, and D+B coherent superpositions**. Our objective is to understand how the various possible combinations of adiabatic states can lead to energy deposition in specific combinations of normal modes. We have studied all the "pairs" of states (plus the full 8-state problem) possibilities in our computations. However, note that the B–B state combination involves an initial equal population of two components while the E–B combination involves three components, etc.). We will be content with a few interesting examples that illustrate the main ideas. For each case, Table 1 gives the allowed linear couplings and electron dynamics for these cases. Thus the normal modes that are stimulated in the first fs will be those identified in Table 1 column 2 (coupling matrix element) while the electron dynamics (column 3) "kicks in" slightly later.

We begin with adiabatic dynamics started on the E state (Fig. 5). The purpose of this computation is mainly to demonstrate that the methodology used has the expected behavior when reaching a conical intersection. This computation can be compared with the MCTDH simulations reported by Galbraith et al.[12] The time dependence of populations of the diabatic hole states (Fig. 2) is shown on the left panel (Fig. 5a) and the normal mode displacements, for the non-totally symmetric modes, are shown on the right (Fig. 5b). We should point out that while states $D_7$ and $D_6$ are degenerate, in practice $D_7$ is 0.0001 $E_h$ higher in energy than $D_6$ and so gets populated in decay from E (i.e., we do not block the problem by symmetry so this energy difference reflects the numerical accuracy) $D_6$ also gets populated but it only becomes significant at around 75–80 fs. This is a consequence of the separation of the degenerate components by subgroups. However, our sole purpose in running this computation is to illustrate that the single state dynamics at a conical

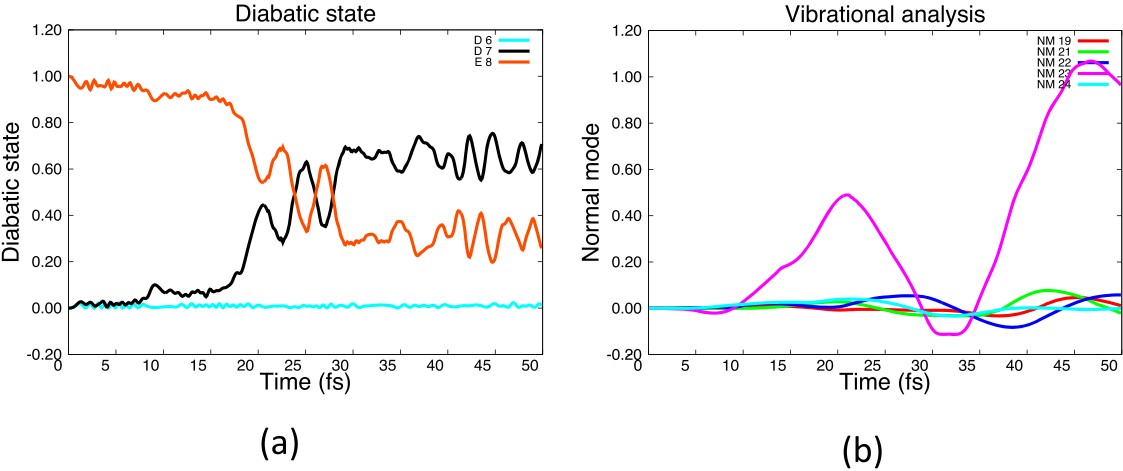

**Fig. 5 Decay of state E at an E/D conical intersection at 10–20 fs with the stimulation of mode 23 (see Table 1).** Note that $D_7$ is populated first while $D_6$ begins to become populated only after 75–85 fs (not shown). **a** Electron dynamics (diabatic populations). **b** Normal mode populations (weighted by gross populations).

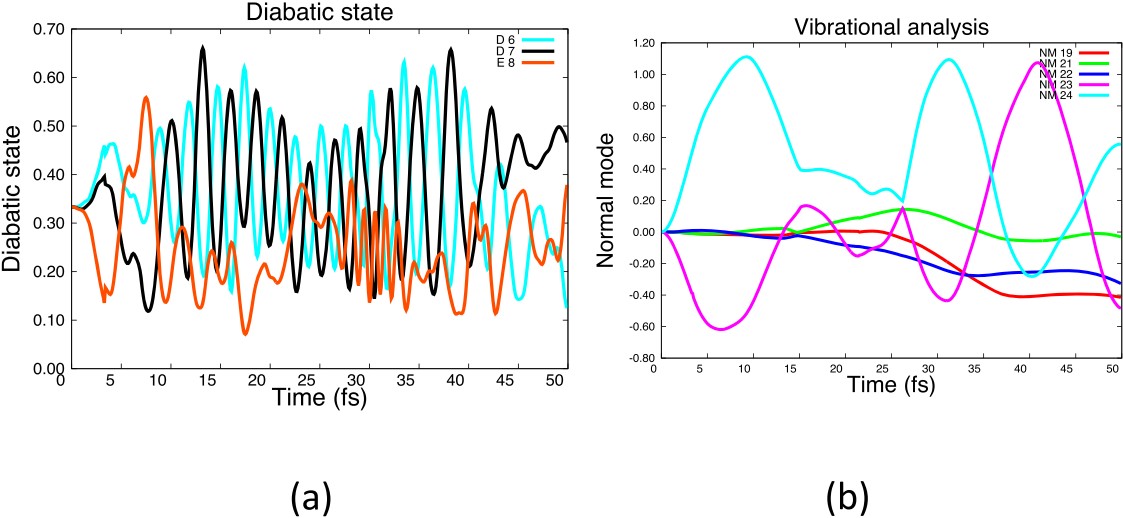

**Fig. 6 Dynamics with an initial superposition of E+D ($D_6$, $D_7$). a** Population of diabatic states. **b** Normal mode populations, weighted by gross population.

intersection obeys the same rules (Table 1), and that electron dynamics is initiated at the conical intersection as expected. We can see that the population starts to decay from the $E_8$ state to $D_7$ after about 7 fs (Fig. 5a) and a significant part of the population is transferred with 20 fs, in agreement with the computations reported by Galbraith et al.[12]. In agreement with the selection rules given in Table 1, mode 23 is stimulated (Fig. 2b), i.e., E + $D_7$ mixing produces a gradient along mode 23.

Now let us consider a different case, with a 3-state coherent superposition of the E (blue) and the two D states ($D_6$ and $D_7$) as shown in Fig. 6. Now, in addition to E→D population transfer (at 10 fs Fig. 6a), we see E–$D_6$ and E–$D_7$ electron dynamics from 10 fs. The E–$D_6$ and E–$D_7$ off-diagonal matrix element (see Table 1) stimulates modes 23 and 24 (Table 1). So the behavior of the case where state E decays to state D (Fig. 5) and the case where one starts with a coherent superposition of E and D (Fig. 6) is slightly different since the full E–$D_6$ motion is in addition to the E–$D_7$ motion. Note that there is also some small stimulation of modes 19, 21, and 22 from the E–D electron dynamics (Table 1, column 3).

Now let us consider the mixing of two nonadjacent states E and B ($B_3$, $B_4$) (Fig. 7) as well as the D+B mixing (Fig. 8). This is the generalization of the E→D and D→B conical intersection pathways considered by Galbraith et al.[12].

For the E+B case (Fig. 7), there is the expected (Table 1 coupling selection rules) stimulation (Fig. 7b) of mode 24 from the $B_3$–$B_4$ electronic coupling (Table 1). However, one also sees stimulation of mode 21 from the E+$B_4$ coupling and modes 19 and 22 from the E+$B_3$ coupling (Table 1). Thus one sees a nonadiabatic coupling between E and B even though the states are well separated in energy. The E+$B_3$ and E+$B_4$ non-adiabatic dynamics is an example of the type of motion normally seen at a (Jahn–Teller) conical intersection (i.e., from derivative coupling between E+$B_3$ and E+$B_4$). There is an additional observation, that in Fig. 7a we see that the intervening $D_6$ state is populated as well. Thus a coherent superposition of states E and B also populates the intervening D state after a few fs.

We now discuss the electron dynamics resulting from E+$B_3$ and E+$B_4$ superposition. This case was given analytically in Fig. 3a, parts i and ii, to give the superposition shown in part iii

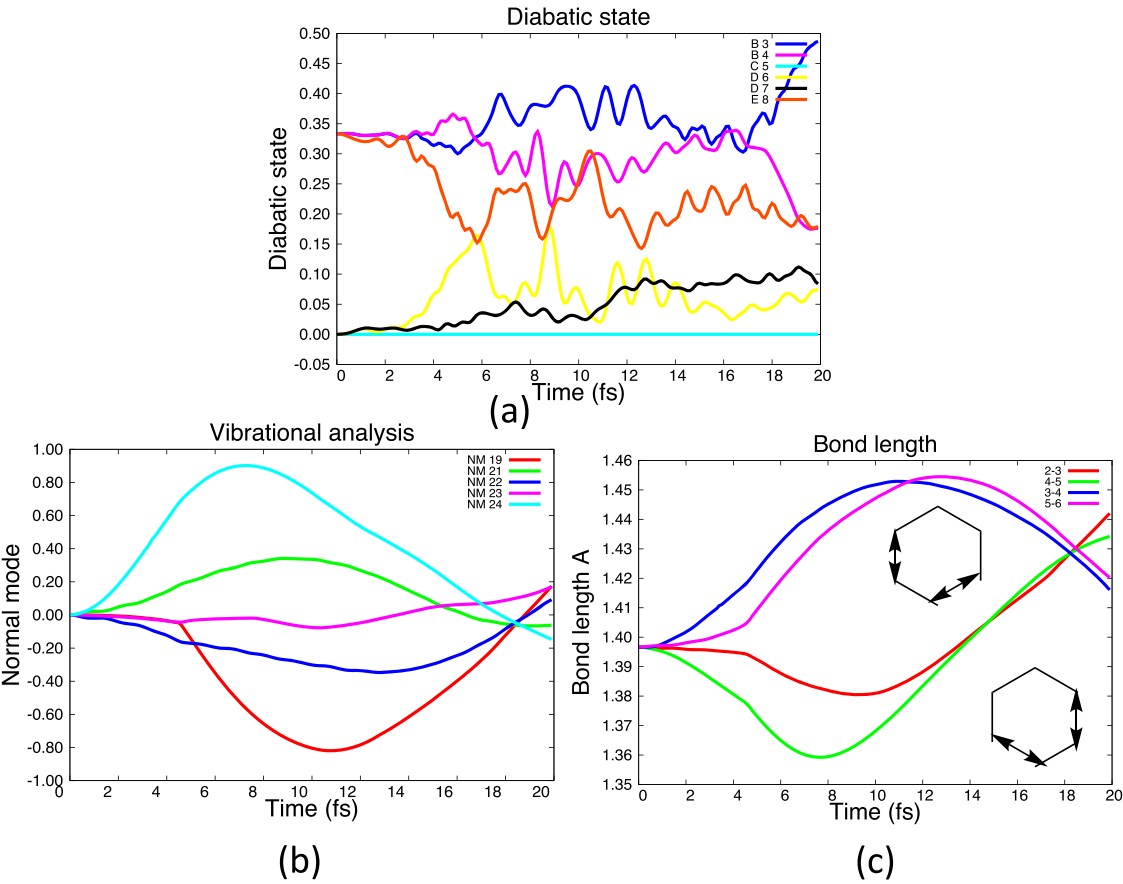

**Fig. 7 Results of E+B, 3-state initial superposition with Qu–Eh dynamics.** The labels $E_8$, D ($D_7 D_6$), etc. refer to the singly ionized states in Fig. 2. The atom labels 1–6 in the carbon atoms start at the top of the hexagon. **a** Populations of diabatic states. **b** Normal mode displacements. **c** Bond vibrations for dominant stretches. (Note that the apparent sudden onset of nm 19 is just an illusion from the plotting software. Examination of the raw data shows that it is continuous).

(see also Fig. 1b). The corresponding superposition of normal modes 23 and 19 gives the nuclear motion shown in Fig. 3a part vi. This motion preserves $C_{2v}$ symmetry, corresponding to fragments (C–C) + (C–C–C–C). The corresponding analysis for E+$B_4$ is given Fig. 3b (parts i, ii, and iii) and shows the superposition of nm 19 and 24 corresponding to (C–C) + (C–C–C–C) fragments. In Fig. 7c we show the two coherent bond stretches present in the overall motion. These clearly show coherent stretchings of $C_3$–$C_4$ plus $C_5$–$C_6$ as well as $C_2$–$C_3$ plus $C_4$–$C_5$. These motions are the precursors to the (C–C) + (C–C–C–C) fragmentation. Thus, while Fig. 7b shows the activation of the normal modes consistent with the symmetry in Table 1, Fig. 7c shows the resultant effect on all the nuclear motion and thus the energy is deposited in the bond stretchings corresponding to the (C–C) + (C–C–C–C) fragmentation orientation. As mentioned previously, we have just taken all the mixings to be positive so there are many equivalent sets of results with different (C–C) + (C–C–C–C) fragmentations that are related by symmetry.

The D+B dynamics (also nonadjacent states) is shown in Fig. 8a, b, and c. Mode 24 is stimulated initially by the B–B and D–D coupling (Table 1) corresponding to B+B and D+D Jahn–Teller dynamics. At a later time, there is a small stimulation of modes 21 then 23 from the electron dynamics (Table 1). This case was also analyzed for the $B_3$–$B_4$ electron dynamics in Fig. 3b (parts iv, v, and vi). There it can be seen that the superposition of nm 23 and 24 from the B+B electron dynamics (Fig. 3b parts iv,

v, and vi correspond to (C–C–C) + (C–C–C) fragmentation and this can be seen in Fig. 8c.

To conclude this discussion, we observe that the population of (i) a coherent superposition of E and B states or (ii) the coherent superposition B and D states results in non-adiabatic dynamics without any consideration of an explicit conical intersection decay. Further, the E+B vs. D+B mixings control the population of the (C–C) + (C–C--C–C) stretchings (Fig. 7d) versus (C–C–C) + (C--C–C) (Fig. 8d), respectively.

Further insight into the E+$B_3$ superposition (Fig. 7a) is obtained by examining the spin density oscillations which provide evidence for electron dynamics in addition to the behavior of the diabatic populations in Fig. 3 (parts i, ii, and iii). This is given in Supplementary Note 5 Figure S3. There one can see that the spin density oscillations are completely consistent with Fig. 3a (parts i–iii). Thus we have additional evidence for the E+B dynamics in the situation where one is not near a conical intersection (i.e., one would expect[18] to see electron dynamics near a conical intersection).

**Conclusions**. In a recent experiment, the non-adiabatic dynamics of a coherent superposition of 8 cationic states of benzene[12] was measured. Supporting theory suggested a decay mechanism involving successive decay E→D and D→B. In this study, we have shown that the non-adiabatic effects (e.g., electron dynamics) associated with E+B coherent superpositions can be seen far

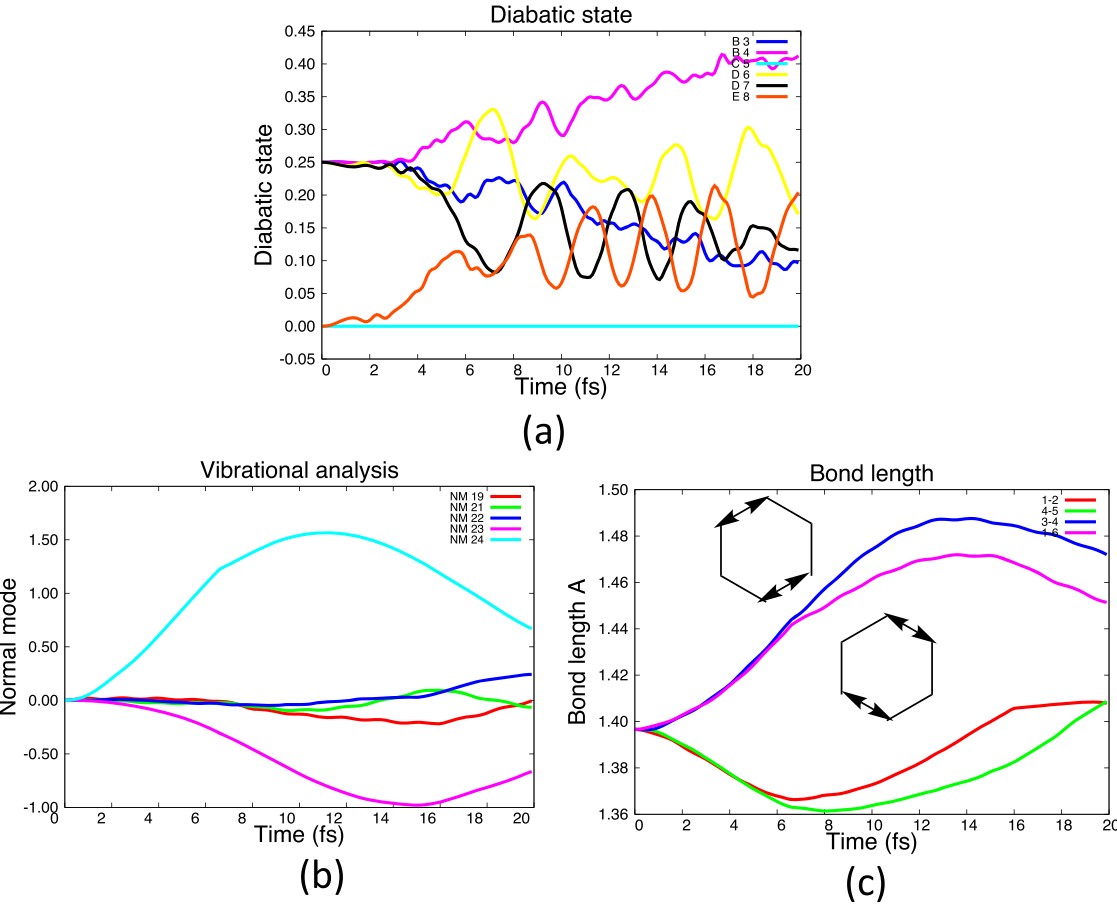

**Fig. 8 Results of D+B, 4-state Qu–Eh dynamics. a** Populations of diabatic states. **b** Normal mode displacements. **c** Bond vibrations for dominant stretches.

from the conical intersection and may be partly responsible for the $C_4$ fragments.

## Methods

All theory and software are documented in refs. [21–26].

## Data availability

Data sharing not applicable to this article as no datasets were generated or analyzed during the current study.

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

## Acknowledgements

The authors are grateful for financial support from Gaussian, Inc. and the UK Engineering and Physical Science Council (grant EP/T006943/1). All computations were carried out at the Imperial College Research Computing Service (DOI: 10.14469/hpc/2232).

## Author contributions

M.A.R. and T.T. performed all the computations reported in this work. G.A.W. supervised T.T. T.T., G.A.W., and M.A.R. all contributed to the drafting of the paper.

## Competing interests

The authors declare no competing interests.
