## [Peer Review File · Communications Chemistry]

Reviewers' comments:

Reviewer #1 (Remarks to the Author):

This is a very interesting paper, which points out some general principles about dynamics of molecules that exist in superpositions of two or more electronic states (using benzene cation as an example). Most importantly, they point out that the symmetry of the force vector may be broken when the molecule populates a linear combination of adiabatic electronic states, resulting in fragmentation. This is a useful concept to present to a broad chemistry audience, and may have important consequences in a range of problems beyond the present benzene experiment. As such, it should be an influential paper. The methods applied are appropriate for the questions being asked. The conclusions follow from the evidence provided. I recommend publication after the authors consider and address the below comments.

There may be a more intuitive way to visualize these symmetry breaking forces that the authors may want to explore. When the molecule exists in a superposition of electronic states, the electronic wave function itself has broken symmetry. Visualizing the superposition hole-state itself for the superpositions described in this work might shed light on the physical basis for the symmetry-breaking force. (For example, perhaps it has bonding lobes on the bonds that will eventually break.) I encourage the authors to plot these states, and if they find that they yield some useful information, include them in the paper.

Also, I wonder about whether the initial conditions (which assume a specific phase between all pairs of hole states) are sufficiently representative. If coherences between electronic states matter, one would expect these phases to matter, and it is not obvious that all possible choices are "equivalent" as claimed on page 3. Some additional discussion/justification is warranted.

Reviewer #2 (Remarks to the Author):

The article entitled "Control of Nuclear Dynamics in the Benzene Cation by Electronic Wavepacket Composition" by T. Tran et al. is a proposal for interpreting the results presented by Galbraith et al; in Nature Communications 8 1018 (2017).

This article showed experimental results in which a neutral benzene molecule was ionized by an ultrashort XUV pulse (10fs) creating excited cations that are subsequently probed by an ultrashort IR pulse.

The probing mechanism is the XUV-IR delay dependent yield of specific fragments created upon IR excitation, revealing the dynamics occurring in the excited cationic benzene.

These results were previously interpreted as the manifestation of the non-adiabatic relaxation of the electronic population through conical intersections.

Here the authors proposed a different theoretical methodology and show that non-adiabatic dynamics might also occur far from the conical intersections and between non adjacent electronic states.

They further identify 2 superposition of electronic states that could lead to the 2 fragmentation pathways discussed in Galbraith et al.

The article is interesting and discusses a non-conventional process that might play a role in XUV induced ultrafast dynamics in molecule systems in general.

I think the community should be aware of this work as it could motivate further experimental investigations.

I have a series of questions and remarks that the authors could address before I can conclude on the relevance of this work:

1- The interpretation proposed in this article relies on the coherent superposition of 8 electronic

states.

In the benzene experiment by Galbraith, the pump process is a photoionization process. I would tend to think that, in itself, the photoionization process does not produce a coherent superposition of electronic states (for instance in the case of ionization by a narrow bandwidth synchrotron radiation).

In order to produce coherent electronic excitation, the bandwidth of the ionizing pulse needs to be broad enough then it covers the energy spacing between the electronic states. I guess this is the requirement to make the states indiscernible upon ionization.

But in this experiment by Galbraith the ionizing pulse is 1eV broad while the 8 states are covering at least 5eV. So I would say that only some coherence between a limited number of states is possible. Is that correct? Can the authors comment on that? How does limited coherence modify the proposed interpretation?

2- In the experiment, the dynamics in the cations is observed by measuring the XUV-IR delay dependent variation of the fragmentation yield. It means that the fragmentation is induced by the IR probe and occurs in the final state of the IR induced cation-cation transition.

In other words, the dynamics occurs in stable benzene cations, the fragmentation only concerns the measurement of the dynamics.

In the theory presented here, the authors make a direct connection between bond elongations of the XUV prepared benzene cation and fragmentation measured in the experiment, induced by IR. I think this connection needs to be discussed further. How do the authors make this connection? Can the author comment on that?

3- The role of nonadiabatic dynamics away from the CIs in these experiments is very interesting. However, I found this interesting message lost over many details in the text. It would be valuable for the reader to have a simple description of the process, as it is usually done for instance when discussing about CI.

For the moment the explanation is given in terms of Hamiltonian (Page 5), but a more "hand waving" explanation giving the essence of the process would certainly have more impact on the community and more striking for experimentalists.

4- The comparison between the simulations and the experimental results is not trivial because one essentially compares a simple exponential decay from which a timescale can be extracted from a simple fit (experiment,) and the complex pattern obtained in the simulation (a superposition of several oscillating features).

It would be helpful to include in figure 4,5 etc... some indications on what to look at, to guide the reader and help to identify which fraction of the figure he/she has to look at.

5- The first obvious consequence of coherent dynamics is the existence of revivals in the populations. This is what is observed in the calculations. Do we expect these oscillations in the experiment too?

They are not observed in Galbraith et al, were they expected in this specific experiment?

some details;

6- In the abstract "The original HHG measurements were supported by MCTDH simulations" I think the term "HHG measurements" would refer to a different type of experiments (namely HHG spectroscopy). To avoid confusion, better use "XUV induced ultrafast dynamics" or something similar.

7- Page 2 "The non-adiabatic dynamics of a coherent superposition of 8 cationic states of benzene was studied experimentally" I don't think the authors of ref 12 claim that the 8 states are coherently excited. I guess this should be modified.

8- Please mind typos that sometimes make the reading difficult. ex: caption figure 5 "23 Pnd 24....

around 20 fA", P17 "the corresponding bond length changes (figure 6D) are created very quickly" etc...

Overall, in my opinion the author should modify the manuscript in order to present their work as a more general interpretation of existing XUV induced dynamics. In that sense the focusing in Galbraith et al might only serve as an illustration but not as a main justification of this work. I think, even without link to Galbraith et al this work would certainly deserve publication and would be of interest for the community.

Although I am not convinced that in the specific case of Galbraith et al.; the proposed interpretation is entirely valid, I still find the article interesting and timely, consequently I could recommend a modified version for publication.

Reviewer #3 (Remarks to the Author):

This is a review of the manuscript "Control of Nuclear Dynamics in the Benzene Cation by Electronic Wavepacket Composition" by authors Tran, Worth, and Robb. The main goal of the study is to give mechanistic insight into coupled electron-nuclear dynamics and to identify which electronic superpositions might lead to which kind of nuclear motion, for the example case of the benzene cation. This topic is potentially interesting for theoretical chemists in the nonadiabatic dynamics field and in ultrafast spectroscopy.

The coupled electron-nuclear dynamics is studied here in an interesting and novel way, by preparing a large number of different initial electronic wave functions and observe its influence on the subsequent nuclear dynamics. They conclude that indeed different electronic linear combinations lead to different nuclear evolution. This is also of potential interest, as it shows that experiments could in principle control to some extent the fragmentation of the benzene cation.

Unfortunately, the manuscript is very difficult to follow. Naturally, benzene is a highly symmetric molecule and therefore requires an extensive treatment of symmetry selection rules. Indeed, the authors dedicated several figures and discussion to the topic of symmetry. However, incomplete or confusing labeling and unintuitive schemes make it difficult to read the essential information from the figures. Furthermore, there are some essential methodical pieces of information missing, which make it difficult to interpret the presented results and assess their correctness. More detailed criticism is given below. As the work can currently not be fully assessed, I can only recommend publication of the manuscript after extensive revision.

Main issues

1) Discussion of the influence of symmetry on the PESs

In the current form, the discussion of symmetry and the potential energy surfaces in the manuscript is probably very hard to follow for readers without a solid background on the previous work on the benzene cation. Here, comprehensive discussions like in Ref 12 (Galbraith et al) and Döscher et al, JCP 2002, 117, 2645 proved very useful. I therefore suggest that the readers are made aware of these previous discussions and that key information is mentioned in a concise way in the manuscript.

Figure 2 is not organized well and the labels are confusing. It appears that the labels on the right refer to the overall D_{6h} symmetry and give the historical spectroscopic state labels (X, B, C, D, E). However, it is not immediately obvious what the numbers are and why the irrep label for state E is missing (state E should be B_{2u} in D_{6h}). Furthermore, the left side the figure introduces additional

irrep labels that refer to D_{2h} symmetry. Here, the labels for the X states are inexplicably missing. Furthermore, some readers might wonder what is the special significance of the D_{2h} symmetry in this work, which is nowhere explained in the manuscript. Why is the D_{2h} subgroup more relevant than other subgroups (in plane: C_{6h}, D_{3h}, C_{2v}(x), C_{2v}(y), C_{2h}(z) D_{2h}; out of plane: C_{6v}, D_{3d}, C_{2h}(x), C_{2h}(y), C_{2v}(z), D₂) that could be reached by different normal mode deformations? Which deformations are responsible for symmetry lowering to D_{2h}? Even if the authors restrict their discussion to modes 13, 19, 21, 22, 23, and 24 (as described below), they should consider symmetry lowering to D_{3h}, C_{2v}(x), C_{2v}(y), C_{2h}(z), and D_{2h} on the same footing.

Figure 3 is probably even more difficult to comprehend than Figure 2. Here, the visualization of normal mode displacements through Lewis structures is extremely unintuitive and untypical. Using simple line drawings with the most important displacements indicated by arrows would be much more descriptive. For example, the arrows can be drawn double-headed, so that each normal mode can be depicted by a single drawing instead of a pair of drawings. This would also allow removing the otherwise superfluous example in Figure 3b. Additionally, it might be advantageous to show the full normal mode depictions (as in b) in a supporting information document.

I suggest to also indicate for each normal mode the associated symmetry lowering and the irreps of the electronic states in the lowered point group. For example, NM 19 is a b_{2u} mode and lowers symmetry to D_{3h}. In D_{3h}, states B₃, B₄, B₆, and D₇ become two pairs of E' states, which couple through the totally symmetric Hamiltonian. This information can probably be easier presented as a table.

Furthermore, the selection criterion of the shown 6 normal modes was never explicitly explained. It can therefore be recommended that in the beginning of the manuscript the connection between the fragmentation of benzene and the 6 C-C stretch modes (a_{1g}+b_{2u}+e_{2g}+e_{1u}) is made. Then it can be discussed which pairs of states these modes couple.

Also note that the entries under "B_{2u}" are probably not correct. B_{2u} modes should couple states B with D, but not states B with E (only E_{1u} modes couple B-E).

In order to remove one possible source of confusion, it is also advisable to change the labels for the normal mode irreps from upper case to lower case (e.g., normal mode 19 is a b_{2u} mode). This is in line with standard convention that normal modes are indicated with lower case letters.

2) Definition of the electronic state basis

It appears that several incompatible electronic state nomenclatures are used. In Figure 2, the electronic states are introduced with their spectroscopic state labels (X, B, C, D, E) and the hole orbitals. Such spectroscopic state labels are typically understood to refer to diabatic states. Also the labels of the population plots in Figures 1, 4, 5, 6, 7 mention that diabatic populations are plotted.

However, in the first manuscript part, the authors claim that "we have taken all the weights of the adiabatic states to be positive", clearly indicating that adiabatic states are used. Combinations of adiabatic states are also mentioned at other places in the manuscript. Furthermore, as the calculations are performed on-the-fly, it is not immediately obvious how the electronic structure calculations would be diabaticized, so it can be assumed that adiabatic states are used. The diabaticization scheme from Ref 14 (orbital localization) is apparently not used, because with orbital localization it is not possible to reach a diabatic basis like the one shown in Figure 2, which has delocalized and symmetry-adapted orbitals.

Thus, it is not possible to properly interpret Figures 1, 4, 5, 6, 7, as it is not clear how the shown "diabatic populations" are obtained and whether they are in fact referring to the same diabatic basis as the orbitals shown in Figure 2.

Relatedly, the authors claim that "A notable feature of QuEh method is that the full derivative coupling is included in the expression for the gradient". This requires more explanation. In Ref 14, a diabatic basis from localized orbitals was used to set all derivative couplings (=nonadiabatic couplings between adiabatic states) equal to zero. In the present work, is this done in a similar way (through localization diabatization), or are all the derivative couplings explicitly treated? Or do you refer to the derivatives of diabatic off-diagonal Hamiltonian matrix elements?

3) Computational details missing

There are several important computational details missing. First, the authors indicate that a CAS-CI method was used for the on-the-fly electronic structure computations. In this case, it is necessary to specify how the orbitals were optimized (e.g., inactive-active rotations). Was this done with a simple HF or SS-CASSCF calculation? Or are the simulations actually based on SA-CASSCF instead of CAS-CI?

Second, the authors claim that "A set of vibrational normal mode coordinates is used as an orthogonal basis." However, according to the description of the QuEh method, direct products of Gaussian wave packets and electronic states are used as basis functions. The vibrational normal modes only span the phase space populated with the Gaussian wave packets.

Third, the authors did not provide details regarding the definition of the normal modes. On which level of theory were those obtained? Were the normal modes properly symmetrized? How were the degenerate normal modes resolved?

Fourth, which normal modes were considered in the simulations? Was the full set of $3N-6$ normal modes included, or were only the most important modes selected?

Fifth, how was the initial momentum distribution, based on 25 gwps, computed? Which normal modes received gwps with "extra" momentum? Was the initial momentum distribution symmetry-adapted?

Sixth, when claiming "We also use a narrower width for the gwp", what is the reference width of the gwps that is compared to?

Seventh, were the computations checked for convergence with respect to the number and distribution of basis functions?

Eighth, what is the "Ehrenfest electronic structure method"? The Ehrenfest method is a method for nonadiabatic dynamics, and requires coupling to a proper electronic structure method (like HF, CASSCF, MRCI, ...). What is meant by "has been interfaced with a development version of Gaussian"? Did the authors intend to write "implemented" instead of "interfaced"?

Minor issues

4) A scheme showing the relative energies between the states and some potential energy curves (like Ref 12 Fig 1) would be very helpful for the readers to get familiar with the system.

5) Figures 1, 4, 5, 6, 7: Please indicate in the caption that panels b and c show position expectation values.

6) On page 5, the authors write "... only if $\alpha^I \times \alpha^{Qi} \times \alpha^{II} = E$ (E is the identity ...)". This is fundamentally wrong. The direct product of several irreps can be decomposed to a direct sum of irreps, but it can not be equated with a symmetry operation. The actual requirement here is that the direct sum contains the totally symmetric irrep.

7) "The gradient of the intrastate terms must always be totally symmetric." This is also imprecise language, and it would be more accurate to write "the intrastate gradient is only non-zero along totally symmetric modes (and for the Jahn-Teller-active e_{2g} modes)."

8) "There are 36 pairs of states" This is not true. With 8 states, there are $8 \times 7 / 2 = 28$ pairs of states. What the authors should explicitly indicate here is that they also tested all 8 cases where all population is initially in one single state.

This statement is especially confusing as the authors do not present a single simulation with non-zero initial population in exactly two states. Figure 1 shows an initial linear combination over 8 states, Figure 4 starts in one state, Figures 5 and 6 in three states, and Figure 7 in four states.

9) With all population initially in the non-degenerate state E, no initial gradient that lowers symmetry, and equivalent couplings E-D₇ and E-D₆, it can be expected that both diabatic D states should behave identically. Can the authors explain the spontaneous symmetry breaking, i.e., that the system spontaneously decides to only populate D₇ instead of the equivalent D₆ state? And likewise, why is only NM 23 excited, but not the equivalent NM 24? Are the initial conditions fully symmetrized? (or are actually adiabatic populations plotted? See comment 2).)

10) Figure 6: Can you explain the unexpected sudden change of momentum for NM 19 at about 4.3fs in panel b? What kind of force can accelerate the molecule so quickly without external influence?

11) Figure 7: The caption is confusing. The figure does not show "D+B₃" but rather a "B₃+B₄+D₆+D₇" linear combination.

12) Figure 8: This figure is very confusing. In (a), it is unclear why a combination of the diagrams on the left (indicating displacements to the top and bottom of the figure) should lead to C₃+C₃ fragmentation in direction top left <-> bottom right.

In (b) could the authors explain the logic of adding the diagrams? Why are "double bonds" sometimes added to yield "triplet bonds" and sometimes not?

What about other dissociation channels? There should be three symmetrically equivalent channels like shown in (a) and three equivalent channels as in (b), for a total of 6 channels.

13) Figure 9: Unfortunately, this figure has very low quality. The axis labeling is insufficient to read the oscillation period. It is not even possible to check whether all plots have the same axis range. Furthermore, here the "benzene pictograms" are used for yet another completely unrelated concept (spin densities). Would it be possible to indicate the spin density oscillations with other schemes, for example by showing red and blue orbital lobes on top of the hexagon?

General Reply to all referees

We are grateful to referees for taking the time to read the paper so thoroughly and for many constructive criticisms and suggestions. With this revision we believe that we have responded to all the points raised.

This response is organized into 2 parts: i) a general response, focussing on the most important changes, (which follows next), indicating how the paper has been revised and ii) a briefer response to each individual point raised by referees.

Accessibility/readability

By removing some of the content related to a comparison with experiment that was (as suggested by referees) rather tenuous in any case, we have used the space to expand the computational details and the conceptual aspects of the electron dynamics of a coherent superposition.

Connection to experimental results: We agree that this connection is somewhat tenuous because there is some doubt as to whether the states excited are truly coherent. We have now discussed this aspect and followed the advice of referee 1 and 3 so that we now present our results in a more general way but “motivated by experiment”

Visualisation of coherent superpositions: Referee 1 has suggested that we should explore the possibility of “visualization” of a superposition of states so we might get some insight as to the physical interpretation of symmetry breaking. It seems to us that attempting to mix the orbitals *per se* could be very difficult to interpret. So we have taken a simpler more visual approach. If one recognizes that both individual C-C sigma bond-orbitals and the individual C-C stretchings are permuted under the operations of D_{6h} , then these 6 functions can be taken as a reducible representation where each irreducible representation ($a_{1g}+b_{2u}+e_{2g}+e_{1u}$) occurs once. Thus the symmetry adapted linear combinations are defined analytically as long as the degenerate representations are restricted to a suitable subgroup (we used D_{2h}). These symmetry adapted linear combinations are in 1:1 correspondence with the orbitals and normal modes used in the computation. Thus we have developed a pictorial representation of the orbitals (figure 2a) and the older version (MO plots) of this figure are now in SI. But it is clear that the symmetry adapted linear combination of these bond functions reproduce the shape and nodal properties of the MO plots (i.e. compare Figure 2a with Figure S1 in SI). The advantage of this representation is that one can compute and visualize any superposition algebraically both for the states and for the normal modes. This fact has enabled us to understand the effect of a superposition in 2 ways 1) the gradient at time zero that comes from the matrix element selection rule and 2) electron dynamics, which gives a “nudge” to the nuclei (not discussed previously) ; see new figure 1b. Since we

can now understand the effect of the electron dynamics in this way, we have moved an improved version of the figures on spin density to SI

Computational details: We have now separated this section under a separate heading and we have included additional detail in SI. S1 initial conditions and the Qu-Eh algorithm

The points and definitions discussed there include. 1) adiabatic states, i.e. diagonalization of H at a given geometry 2)Diabatic states: these are CSF associated initially with adiabatic states at time zero (Because the orbitals change with time these "reference hole states" change slightly with time) 3) definition of the cas space: CAS 15 electrons 8 orbitals 4) Discussion of electronic and nuclear dynamics including the electrons (see SI S2 for details of initial conditions) : TSE Ehrenfest CAS with orbitals propagated to second order) 5) Normal modes (from 6-31g* computation on neutral ground state degenerate rep correspond to D_{2h}) 6) gwp: The gwp were built from 12 normal modes (corresponding to 6C-C stretches and 6 C-C-C bends) corresponding 25 wp from normal modes (one test run with all addition nm but classical propagation (no change in results). 7) discussion of Ehrenfest gradient (ref 15). Here we emphasise that there are off diagonal terms (called derivative couplings in other contexts) that must be included as well as cp-mcscf terms 8)Initial conditions briefly mentioned but also fully detailed in SI In particular 9) we discuss the momentum distribution ii)width of gwp 10)Subgroup chosen only to define degenerate irrep. No loss in generality 11) initial phases of superposition

Reply to referees in red

Reviewer #1 (Remarks to the Author):

This is a very interesting paper, which points out some general principles about dynamics of molecules that exist in superpositions of two or more electronic states (using benzene cation as an example). Most importantly, they point out that the symmetry of the force vector may be broken when the molecule populates a linear combination of adiabatic electronic states, resulting in fragmentation. This is a useful concept to present to a broad chemistry audience, and may have important consequences in a range of problems beyond the present benzene experiment. As such, it should be an influential paper. The methods applied are appropriate for the questions being asked. The conclusions follow from the evidence provided. I recommend publication after the authors consider and address the below comments.

There may be a more intuitive way to visualize these symmetry breaking forces that the authors may want to explore. When the molecule exists in a superposition of electronic

states, the electronic wave function itself has broken symmetry. Visualizing the superposition hole-state itself for the superpositions described in this work might shed light on the physical basis for the symmetry-breaking force. (For example, perhaps it has bonding lobes on the bonds that will eventually break.) I encourage the authors to plot these states, and if they find that they yield some useful information, include them in the paper.

Since the 6 C-C bonds and the 6 C-C stretches form a reducible permutation $\{h_1...h_6\}$ representation of the D_{6h} point group, it is possible to give a pictorial (and algebraic) representation of both the normal modes and the MO as linear combinations of the basis $\{h_1...h_6\}$. This algebraic representation permits one to have a rough visualization of a superposition of either the normal modes or the electronic states. This information proves to be a useful adjunct and has been added to the paper. Indeed since the reduction of this permutation contains each irrep. only once, this analysis has replaced the MO diagrams in figure 2. As we show later, these “diagrams” can be literally added to give the nodal pattern of the pairwise superpositions. We feel that these simplified representations are much easier to follow than the MO plots themselves and these have now been moved to SI figure S1.

We have now created a new figure (figure 1) that gives a mechanistic overview using this representation. Here, and in figure 3 one can see that the formation of a superposition has the effect of localization. This is the origin of the electron dynamics between the state E and B where the electron dynamics exchanges the positive and negative lobes with time.

Also, I wonder about whether the initial conditions (which assume a specific phase between all pairs of hole states) are sufficiently representative. If coherences between electronic states matter, one would expect these phases to matter, and it is not obvious that all possible choices are “equivalent” as claimed on page 3. Some additional discussion/justification is warranted.

We have now discussed this in the computational details. For a pair of states it is clear the phase mixing just changes the direction of the electron dynamics (e.g. figure 1b or figure 3). For different phases of mixing, there will be several equivalent (by symmetry) c3+c3 fragmentation patterns. Changing the phase between states in the superposition just generates equivalent results related by symmetry. This is now discussed in text.

\

Reviewer #2 (Remarks to the Author)

:

The article entitled "Control of Nuclear Dynamics in the Benzene Cation by Electronic Wavepacket Composition" by T. Tran et al. is a proposal for interpreting the results presented by Galbraith et al; in Nature Communications 8 1018 (2017). This article showed experimental results in which a neutral benzene molecule was ionized by an ultrashort XUV pulse (10fs) creating excited cations that are subsequently probed by an ultrashort IR pulse.

The probing mechanism is the XUV-IR delay dependent yield of specific fragments created upon IR excitation, revealing the dynamics occurring in the excited cationic benzene.

These results were previously interpreted as the manifestation of the non-adiabatic relaxation of the electronic population through conical intersections. Here the authors proposed a different theoretical methodology and show that non-adiabatic dynamics might also occur far from the conical intersections and between non adjacent electronic states.

They further identify 2 superposition of electronic states that could lead to the 2 fragmentation pathways discussed in Galbraith et al.

The article is interesting and discusses a non-conventional process that might play a role in XUV induced ultrafast dynamics in molecule systems in general. I think the community should be aware of this work as it could motivate further experimental investigations.

I have a series of questions and remarks that the authors could address before I can conclude on the relevance of this work:

1- The interpretation proposed in this article relies on the coherent superposition of 8 electronic states.

In the benzene experiment by Galbraith, the pump process is a photoionization process. I would tend to think that, in itself, the photoionization process does not produce a coherent superposition of electronic states (for instant in the case of ionization by a narrow bandwidth synchrotron radiation).

In order to produce coherent electronic excitation, the bandwidth of the ionizing pulse needs to be broad enough then it covers the energy spacing between the electronic states. I guess this is the requirement to make the states indiscernible upon ionization. But in this experiment by Galbraith the ionizing pulse is 1eV broad while the 8 states are covering at least 5eV. So I would say that only some coherence between a limited

number of states is possible. Is that correct? Can the authors comment on that? How does limited coherence modifies the proposed interpretation?

In the Galbriath paper they use the 9th harmonic with about 16 eV energy and then a 1 eV probe pulse. We agree that all the states might not all be coherently excited. Further we cannot include the details of the initial excitation and we use a “sudden” approximation. We now point this out more clearly in the text. We agree with the point made at the end of this review “it is probably better to change the thrust of our work as a more general presentation “motivated” by experiment”

2- In the experiment, the dynamics in the cations is observed by measuring the XUV-IR delay dependent variation of the fragmentation yield. It means that the fragmentation is induced by the IR probe and occurs in the final state of the IR induced cation-cation transition.

In other words, the dynamics occurs in stable benzene cations, the fragmentation only concerns the measurement of the dynamics.

In the theory presented here, the authors make a direct connection between bond elongations of the XUV prepared benzene cation and fragmentation measured in the experiment, induced by IR.

I think this connection needs to be discussed further. How do the authors make this connection? Can the author comment on that?

We agree that our work is concerned only with the initial population of normal modes and stretching patterns. As such these patterns may provide the orientation or entropic effect associated with long-term fragmentation. We have strengthened the discussion of this point in the text.

3- The role of nonadiabatic dynamics away from the CIs in these experiments is very interesting.

However, I found this interesting message lost over many details in the text. It would be valuable for the reader to have a simple description of the process, as it is usually done for instance when discussing about CI.

For the moment the explanation is given in terms of Hamiltonian (Page 5), but a more “hand waving” explanation giving the essence of the process would certainly have more impact on the community and more striking for experimentalists.

We have now added such a discussion both in the introduction (figure 1) in addition to the symmetry argument indicated above. Further, there is a new thread in the paper: The role of the electron dynamics (figure 1b and figure 3)

4- The comparison between the simulations and the experimental results is not trivial because one essentially compares a simple exponential decay from which a timescale can be extracted from a simple fit (experiment,) and the complex pattern obtained in the simulation (a superposition of several oscillating features).

It would be helpful to include in figure 4,5 etc... some indications on what to look at, to guide the reader and help to identify which fraction of the figure he/she has to look at.

We have tried to improve both the text and the captions to improve readability here.

5- The first obvious consequence of coherent dynamics is the existence of revivals in the populations. This is what is observed in the calculations. Do we expect these oscillations in the experiment too?

They are not observed in Galbraith et al, were they expected in this specific experiment?

The electron dynamics is very fast 1-2 fs (see section on spin density in SI!) much faster than the oscillations in the normal modes. So it may not be possible to observe the electron dynamics unless a time resolved esr experiment is possible. We have now improved the discussion of the spin density but moved it to SI to keep within the length restriction.

6- In the abstract "The original HHG measurements were supported by MCTDH simulations" I think the term "HHG measurements" would refer to a different type of experiments (namely HHG spectroscopy). To avoid confusion, better use "XUV induced ultrafast dynamics" or something similar.

Fixed

7- Page2 "The non-adiabatic dynamics of a coherent superposition of 8 cationic states of benzene was studied experimentally" I don't think the authors of ref 12 claim that the 8 states are coherently excited. I guess this should be modified.

See discussion above. This statement is now clarified in the text in a new subsection headed computational details

8- Please mind typos that sometimes make the reading difficult. ex: caption figure 5 "23

Pnd 24.... around 20 fA", P17 "the corresponding bond length changes (figure 6D) are created very quickly" etc...

Fixed..

Overall, in my opinion the author should modify the manuscript in order to present their work as a more general interpretation of existing XUV induced dynamics. In that sense the focusing in Galbraith et al might only serve as an illustration but not as a main justification of this work.

I think, even without link to Galbraith et al this work would certainly deserve publication and would be of interest for the community.

Although I am not convinced that in the specific case of Galbraith et al.; the proposed interpretation is entirely valid, I still find the article interesting and timely, consequently I could recommend a modified version for publication.

Agreed: We have changed the thrust of the MS accordingly

Reviewer #3 (Remarks to the Author):

This is a review of the manuscript "Control of Nuclear Dynamics in the Benzene Cation by Electronic Wavepacket Composition" by authors Tran, Worth, and Robb. The main goal of the study is to give mechanistic insight into coupled electron-nuclear dynamics and to identify which electronic superpositions might lead to which kind of nuclear motion, for the example case of the benzene cation. This topic is potentially interesting for theoretical chemists in the nonadiabatic dynamics field and in ultrafast spectroscopy.

The coupled electron-nuclear dynamics is studied here in an interesting and novel way, by preparing a large number of different initial electronic wave functions and observe its influence on the subsequent nuclear dynamics. They conclude that indeed different electronic linear combinations lead to different nuclear evolution. This is also of potential interest, as it shows that experiments could in principle control to some extent the fragmentation of the benzene cation.

Unfortunately, the manuscript is very difficult to follow. Naturally, benzene is a highly symmetric molecule and therefore requires an extensive treatment of symmetry selection rules. Indeed, the authors dedicated several figures and discussion to the topic of symmetry. However, incomplete or confusing labeling and unintuitive schemes make

it difficult to read the essential information from the figures. Furthermore, there are some essential methodical pieces of information missing, which make it difficult to interpret the presented results and assess their correctness. More detailed criticism is given below. As the work can currently not be fully assessed, I can only recommend publication of the manuscript after extensive revision

Main issues

1) Discussion of the influence of symmetry on the PESs

In the current form, the discussion of symmetry and the potential energy surfaces in the manuscript is probably very hard to follow for readers without a solid background on the previous work on the benzene cation. Here, comprehensive discussions like in Ref 12 (Galbraith et al) and Döscher et al, JCP 2002, 117, 2645 proved very useful. I therefore suggest that the readers are made aware of these previous discussions and that key information is mentioned in a concise way in the manuscript.

Figure 2 is not organized well and the labels are confusing. It appears that the labels on the right refer to the overall D_{6h} symmetry and give the historical spectroscopic state labels (X, B, C, D, E). However, it is not immediately obvious what the numbers are and why the irrep label for state E is missing (state E should be B_{2u} in D_{6h}). Furthermore, the left side the figure introduces additional irrep labels that refer to D_{2h} symmetry. Here, the labels for the X states are inexplicably missing. Furthermore, some readers might wonder what is the special significance of the D_{2h} symmetry in this work, which is nowhere explained in the manuscript. Why is the D_{2h} subgroup more relevant than other subgroups (in plane: C_{6h} , D_{3h} , $C_{2v}(x)$, $C_{2v}(y)$, $C_{2h}(z)$ D_{2h} ; out of plane: C_{6v} , D_{3d} , $C_{2h}(x)$, $C_{2h}(y)$, $C_{2v}(z)$, D_2) that could be reached by different normal mode deformations? Which deformations are responsible for symmetry lowering to D_{2h} ? Even if the authors restrict their discussion to modes 3, 19, 21, 22, 23, and 24 (as described below), they should consider symmetry lowering to D_{3h} , $C_{2v}(x)$, $C_{2v}(y)$, $C_{2h}(z)$, and D_{2h} on the same footing.

We have now included a discussion of this in the computational details section. Further we have a new section on symmetry normal modes etc including a discussion on the electron dynamics

Briefly, one needs to make a choice of a suitable subgroup in order to define the orbitals and the normal modes for the degenerate irreducible representations. Any unitary

transformation of the orbitals or the normal modes of a degenerate representation can be used. Requiring that the orbitals and normal modes also be irreducible representations of a subgroup merely chooses a specific unitary transformation. This does not introduce any lack of generality since almost any choice would do. But one must make one choice from the outset. The computed normal modes form the basis for the nuclear dynamics and since one fragmentation pattern was D_{2h} this was the choice that was made.

Figure 3 is probably even more difficult to comprehend than Figure 2. Here, the visualization of normal mode displacements through Lewis structures is extremely unintuitive and untypical. Using simple line drawings with the most important displacements indicated by arrows would be much more descriptive. For example, the arrows can be drawn double-headed, so that each normal mode can be depicted by a single drawing instead of a pair of drawings. This would also allow removing the otherwise superfluous example in Figure 3b. Additionally, it might be advantageous to show the full normal mode depictions (as in b) in a supporting information document.

We have now collected all the symmetry ideas into a unified subsection.

Specific points:

1) figure 2 and 3 have been replaced and unified with a discussion of the permutation representation that can be used both for the normal modes and the electronic states. There are new figures 2 and 3

2) Sub-groups: This is now fully discussed in the computational details

I suggest to also indicate for each normal mode the associated symmetry lowering and the irreps of the electronic states in the lowered point group. For example, NM 19 is a b_{2u} mode and lowers symmetry to D_{3h} . In D_{3h} , states B3, B4, B6, and D7 become two pairs of E' states, which couple through the totally symmetric Hamiltonian. This information can probably be easier presented as a table.

This is a good point and the information has been added to Figure 2b. We have also pointed out that the couplings discussed with respect to the initial gradient are generalized after the initial distortion.

Furthermore, the selection criterion of the shown 6 normal modes was never explicitly explained.

This is now made clear in computational details. We chose the NM that corresponded to the 6 C-C stretches and 6 C-C-C angle bends. We did run one 8 state computation with these modes plus all the remaining modes (treated classically as a reality check)

It can therefore be recommended that in the beginning of the manuscript the connection between the fragmentation of benzene and the 6 C-C stretch modes (a1g+b2u+e2g+e1u) is made. Then it can be discussed which pairs of states these modes couple.

The stretching modes are now collected in a new figure 2b in a conventional way. The coupling of normal modes and electron dynamics is now discussed in figure 1b and figure 3.

Also note that the entries under "B2u" are probably not correct. B2u modes should couple states B with D, but not states B with E (only E1u modes couple B-E).

This figure has been corrected and a simpler version included in supplementary information. We have added table 1 to collect the most important data for the cases considered explicitly in subsequent discussion.

In order to remove one possible source of confusion, it is also advisable to change the labels for the normal mode irreps from upper case to lower case (e.g., normal mode 19 is a b2u mode). This is in line with standard convention that normal modes are indicated with lower case letters.

Done

2) Definition of the electronic state basis

It appears that several incompatible electronic state nomenclatures are used. In Figure 2, the electronic states are introduced with their spectroscopic state labels (X, B, C, D, E) and the hole orbitals. Such spectroscopic state labels are typically understood to refer to diabatic states. Also the labels of the population plots in Figures 1, 4, 5, 6, 7 mention that diabatic populations are plotted.

However, in the first manuscript part, the authors claim that "we have taken all the weights of the adiabatic states to be positive", clearly indicating that adiabatic states are used. Combinations of adiabatic states are also mentioned at other places in the manuscript. Furthermore, as the calculations are performed on-the-fly, it is not

immediately obvious how the electronic structure calculations would be diabaticized, so it can be assumed that adiabatic states are used. The diabaticization scheme from Ref 14 (orbital localization) is apparently not used, because with orbital localization it is not possible to reach a diabatic basis like the one shown in Figure 2, which has delocalized and symmetry-adapted orbitals.

This has now also been clarified in computational details: "Such spectroscopic state labels are typically understood to refer to diabatic states". These are in fact what we have plotted as diabatic states. We have now defined these precisely in the text.

Thus, it is not possible to properly interpret Figures 1, 4, 5, 6, 7, as it is not clear how the shown "diabatic populations" are obtained and whether they are in fact referring to the same diabatic basis as the orbitals shown in Figure 2.

See previous reply. In fact we have tried to discuss this aspect more thoroughly as requested by another referee.

Relatedly, the authors claim that "A notable feature of QuEh method is that the full derivative coupling is included in the expression for the gradient". This requires more explanation. In Ref 14, a diabatic basis from localized orbitals was used to set all derivative couplings (=nonadiabatic couplings between adiabatic states) equal to zero. In the present work, is this done in a similar way (through localization diabaticization), or are all the derivative couplings explicitly treated? Or do you refer to the derivatives of diabatic off-diagonal Hamiltonian matrix elements?

This is also now treated in computational details. The salient point is that the Ehrenfest wavefunction is a linear combination of adiabatic states. The crucial point is to compute the gradient correctly. It is at this point that all the "off-diagonal gradients" between adiabatic states (referred to as derivative couplings) must be computed correctly including the coupled perturbed corrections.

3) Computational details missing

There are several important computational details missing. First, the authors indicate that a CAS-CI method was used for the on-the-fly electronic structure computations. In this case, it is necessary to specify how the orbitals were optimized (e.g., inactive-active rotations). Was this done with a simple HF or SS-CASSCF calculation? Or are the simulations actually based on SA-CASSCF instead of CAS-CI?

This is now discussed in computational details. The orbitals used were started from an 8-state CAS-SCF computation. The propagation of the CI coefficients was carried out in the Ehrenfest fashion. At each step of the Ehrenfest propagation, orbitals were propagated to second order. So it is a form of CAS-CI because the orbitals are not fully optimized. We have put a more complete discussion in SI

Second, the authors claim that "A set of vibrational normal mode coordinates is used as an orthogonal basis." However, according to the description of the QuEh method, direct products of Gaussian wave packets and electronic states are used as basis functions. The vibrational normal modes only span the phase space populated with the Gaussian wave packets.

"direct products of Gaussian wave packets and electronic states are used as basis functions. The vibrational normal modes only span the phase space populated with the Gaussian wave packets" This is now stated more precisely and full initial conditions are given in SI.

"

Third, the authors did not provide details regarding the definition of the normal modes. On which level of theory were those obtained? Were the normal modes properly symmetrized? How were the degenerate normal modes resolved?

Now stated in comp details section: The normal modes we obtained from an analytical frequency computation at the B3Lyp level on the neutral system. Both degenerate orbitals and degenerate normal modes were adapted to D2h

Fourth, which normal modes were considered in the simulations? Was the full set of $3N-6$ normal modes included, or were only the most important modes selected?

We used a subset 12 normal modes spanning the C-C-Cbends and C-C stretches. But a test was done with all the additional normal modes treated classically.

Fifth, how was the initial momentum distribution, based on 25 gwps, computed? Which normal modes received gwps with "extra" momentum? Was the initial momentum distribution symmetry-adapted?

This now discussed in more detail in SI as well as in computational details.

Sixth, when claiming "We also use a narrower width for the gwp", what is the reference width of the gwps that is compared to?

.1 rather than .707 (harmonic oscillator) now discussed in text and SI

Seventh, were the computations checked for convergence with respect to the number and distribution of basis functions?

We now give full details for the dynamics part in SI. We did run one computation with all additional nm treated classical. We ran full shell computations with the 12 NM to give $2 \times 12 + 1$ gwp. This was at the limit of our computational ability. We have now mentioned this in computational details.

Eighth, what is the "Ehrenfest electronic structure method"? The Ehrenfest method is a method for nonadiabatic dynamics, and requires coupling to a proper electronic structure method (like HF, CASSCF, MRCI, ...). What is meant by "has been interfaced with a development version of Gaussian"? Did the authors intend to write "implemented" instead of "interfaced"?

Also clarified in computational details and Yes we should have used implemented as described in reference 15. In addition we now give some details of the algorithm in SI.

Minor issues

4) A scheme showing the relative energies between the states and some potential energy curves (like Ref 12 Fig 1) would be very helpful for the readers to get familiar with the system.

We have now produced a scheme that unifies the sequential potential surface crossings figure 1b vs coherent superpositions (figure 1a)

5) Figures 1, 4, 5, 6, 7: Please indicate in the caption that panels b and c show position expectation values.

Done

6) On page 5, the authors write "... only if α^I times α^{Qi} times $\alpha^{II}=E$ (E is

the identity ...). This is fundamentally wrong. The direct product of several irreps can be decomposed to a direct sum of irreps, but it can not be equated with a symmetry operation. The actual requirement here is that the direct sum contains the totally symmetric irrep.

Of course, absolutely correct now fixed.

7) "The gradient of the intrastate terms must always be totally symmetric." This is also imprecise language, and it would be more accurate to write "the intrastate gradient is only non-zero along totally symmetric modes (and for the Jahn-Teller-active e_{2g} modes)."

fixed

8) "There are 36 pairs of states" This is not true. With 8 states, there are $8 \times 7 / 2 = 28$ pairs of states. What the authors should explicitly indicate here is that they also tested all 8 cases where all population is initially in one single state. This statement is especially confusing as the authors do not present a single simulation with non-zero initial population in exactly two states. Figure 1 shows an initial linear combination over 8 states, Figure 4 starts in one state, Figures 5 and 6 in three states, and Figure 7 in four states.

We have corrected all these things. We have now made this discussion more precise and correct.

9) With all population initially in the non-degenerate state E, no initial gradient that lowers symmetry, and equivalent couplings E-D7 and E-D6, it can be expected that both diabatic D states should behave identically. Can the authors explain the spontaneous symmetry breaking, i.e., that the system spontaneously decides to only populate D7 instead of the equivalent D6 state? And likewise, why is only NM 23 excited, but not the equivalent NM 24? Are the initial conditions fully symmetrized? (or are actually adiabatic populations plotted? See comment 2).)

Symmetry is not imposed on the problem at any stage. So for example the energies of states D6 and D7 differs from D6 by .0001 Hartree. Thus starting from the state E the D 7 state will be populated first. Near the E/D7 conical intersection we have a stimulation of nm23. This is now clarified in the text.

10) Figure 6: Can you explain the unexpected sudden change of momentum for NM 19 at about 4.3fs in panel b? What kind of force can accelerate the molecule so quickly without external influence?

This is just a question of the plotting resolution. We have checked the raw data and there is no discontinuity,

11) Figure 7: The caption is confusing. The figure does not show "D+B3" but rather a "B3+B4+D6+D7" linear combination.

Just a typo. Now corrected

12) Figure 8: This figure is very confusing. In (a), it is unclear why a combination of the diagrams on the left (indicating displacements to the top and bottom of the figure) should lead to C3+C3 fragmentation in direction top left <-> bottom right. In (b) could the authors explain the logic of adding the diagrams? Why are "double bonds" sometimes added to yield "triplet bonds" and sometimes not? What about other dissociation channels? There should be three symmetrically equivalent channels like shown in (a) and three equivalent channels as in (b), for a total of 6 channels.

This figure (figure 8) has been completely redrawn (now figure 2a) using the permutation basis introduced for the orbitals

13) Figure 9: Unfortunately, this figure has very low quality. The axis labeling is insufficient to read the oscillation period. It is not even possible to check whether all plots have the same axis range. Furthermore, here the "benzene pictograms" are used for yet another completely unrelated concept (spin densities). Would it be possible to indicate the spin density oscillations with other schemes, for example by showing red and blue orbital lobes on top of the hexagon?

The full plots of the spin density are now included in supplementary information and the discussion of electron dynamics is included in the main text. (since the permutation basis makes the nature of the electron dynamics clear). We would be over the space restrictions if we put this back in the main text.

REVIEWERS' COMMENTS:

Reviewer #1 (Remarks to the Author):

This paper is ready for publication.

Reviewer #2 (Remarks to the Author):

The authors have addressed my questions and comments.

I find the new version of the manuscript of high interest for the ultrafast community and I therefore recommend its publication.

Franck LEPINE

Reviewer #3 (Remarks to the Author):

In their revised version of the manuscript "Control of Nuclear Dynamics in the Benzene Cation by Electronic Wavepacket Composition" the authors Tran, Worth, and Robb address the many referee comments in a reasonable way and improve the manuscript significantly. I have only a few additional comments, as discussed below. I suggest that these comments are addressed before the manuscript can be accepted for publication.

1) In the supporting information, it is written that "The Qu-Eh method solves the time-dependent Schrödinger equation using a variational solution for both electrons and nuclei." Could the authors please discuss this in slightly more detail? How can the method be fully variational for electrons and nuclei if the orbitals are not optimized between time steps?

2) Also in the supporting information, they write "The molecular wavefunction (see equation 1 in main text) is combined with the nuclear wavefunction ...". Equation 1 in the main text clearly refers to only the electronic part of the wave function ("Now we turn to the electron dynamics."). Hence, please check whether "molecular wavefunction" is the correct term to use in the supporting information.

3) In the rebuttal letter, the authors write "This is now discussed in computational details. The orbitals used were started from an 8-state CAS-SCF computation." I checked the entire manuscript and supporting information, but I could not find this information in the text. Please add to the manuscript that the orbitals were taken from an 8-state CASSCF (presumably at the symmetric D_{6h} minimum geometry?). Please verify that all relevant information given in the rebuttal is included in the manuscript to be made available to the readers.

4) In the supporting information, the authors describe the initial 25 gwp's. Please also add to this section how much initial momentum the gwp's are given.

5) In Figure 1a, the E state should be of B_{2u} symmetry, not A_{2u} symmetry. Please check the entire text for consistency regarding the labels for symmetries, states, modes, and others.

6) Page 7: "Changing these phases merely produces an "equivalent" fragmentation pattern (related to the one obtained by a symmetry)" While I agree that this is true for mixtures of two states, the situation is surely more complicated if more than two states are mixed. Could the authors please discuss this, mentioning that this would lead to additional superpositions that are not considered in this manuscript?

Reply to referee 3 (in red changed text in blue)

Reviewer #3 (Remarks to the Author):

In their revised version of the manuscript "Control of Nuclear Dynamics in the Benzene Cation by Electronic Wavepacket Composition" the authors Tran, Worth, and Robb address the many referee comments in a reasonable way and improve the manuscript significantly. I have only a few additional comments, as discussed below. I suggest that these comments are addressed before the manuscript can be accepted for publication.

1) In the supporting information, it is written that "The Qu-Eh method solves the time-dependent Schrödinger equation using a variational solution for both electrons and nuclei." Could the authors please discuss this in slightly more detail? How can the method be fully variational for electrons and nuclei if the orbitals are not optimized between time steps?

The referee is correct. We have clarified in SI.

.....solves the time-dependent Schrödinger equation using a variational solution for both electrons and nuclei"

replaced by

The Qu-Eh method solves the time-dependent Schrödinger equation for both CI coefficients and nuclear wavefunction expansion coefficients. The equations, obtained by applying the Dirac–Frenkel variational principle, are variational (Theor Chem Acc (2016) 135:187 DOI 10.1007/s00214-016-1937-2). However, the orbitals are only propagated to second order.

2) Also in the supporting information, they write "The molecular wavefunction (see equation 1 in main text) is combined with the nuclear wavefunction ...". Equation 1 in the main text clearly refers to only the electronic part of the wave function ("Now we turn to the electron dynamics."). Hence, please check whether

"molecular wavefunction" is the correct term to use in the supporting information.

Fixed text now reads

The **electronic wavefunction** (see equation 1 in main text) is combined with the nuclear wavefunction which is

3) In the rebuttal letter, the authors write "This is now discussed in computational details. The orbitals used were started from an 8-state CAS-SCF computation." I checked the entire manuscript and supporting information, but I could not find this information in the text. Please add to the manuscript that the orbitals were taken from an 8-state CASSCF (presumably at the symmetric D_{6h} minimum geometry?).

Done! The text now reads as follows.

Here we used a CAS space with 15 electrons and 8 orbitals. **The starting orbitals were taken from an 8-state CASSCF at the symmetric D_{6h} minimum geometry.**

Please verify that all relevant information given in the rebuttal is included in the manuscript to be made available to the readers.

We have checked all of this again!

4) In the supporting information, the authors describe the initial 25 gwp's. Please also add to this section how much initial momentum the gwp's are given.

We have added the following sentence which gives all the initial conditions of the gwp.

In our computations, the expectation value of the momentum for each normal mode, $\langle p \rangle$ is zero. However, each individual gwp has a momentum and each gwp is initially associated with a normal mode. There are 2 gwp associated with related each vibrational coordinate, one with $+p$ and one $-p$ where $p = 11.77$. Note that p is related to the dimensionless mass-frequency scaled normal coordinates and has units of inverse time. The value is chosen so that the overlap with the central gwp is 0.8. The value of p for each gwp is identical and the initial momentum for gwp 1 is 0.

5) In Figure 1a, the E state should be of B_{2u} symmetry, not A_{2u} symmetry. Please check the entire text for consistency regarding the labels for symmetries,

states, modes, and others.

This is fixed and all others checked

6) Page 7: "Changing these phases merely produces an "equivalent" fragmentation pattern (related to the one obtained by a symmetry)" While I agree that this is true for mixtures of two states, the situation is surely more complicated if more than two states are mixed. Could the authors please discuss this, mentioning that this would lead to additional superpositions that are not considered in this manuscript?

We have changed this sentence to read as follows

Changing these phases, and/or the phases of the adiabatic states and of normal modes themselves produces additional superpositions that are not considered in this manuscript.

We have also changed the corresponding discussion in the section : "8-state non-adiabatic dynamics" end of first paragraph, In the same way.